# A Retrospective Analysis to Estimate the Burden of Invasive Pneumococcal Disease and Non-Invasive Pneumonia in Children <15 Years of Age in the Veneto Region, Italy

**DOI:** 10.3390/children9050657

**Published:** 2022-05-03

**Authors:** Elisa Barbieri, Gloria Porcu, Tianyan Hu, Tanaz Petigara, Francesca Senese, Gian Marco Prandi, Antonio Scamarcia, Luigi Cantarutti, Anna Cantarutti, Carlo Giaquinto

**Affiliations:** 1Division of Paediatric Infectious Diseases, Department of Women’s and Children’s Health, University of Padova, 35128 Padua, Italy; carlo.giaquinto@unipd.it; 2Unit of Biostatistics Epidemiology and Public Health, Department of Statistics and Quantitative Methods, University of Milano-Bicocca, 20126 Milan, Italy; anna.cantarutti@unimib.it; 3National Centre for Healthcare Research and Pharmacoepidemiology, Department of Statistics and Quantitative Methods, University of Milano-Bicocca, 20126 Milan, Italy; 4Merck & Co. Inc., Kenilworth, NJ 07033, USA; hutianyan@gmail.com (T.H.); tanaz.petigara@merck.com (T.P.); 5MSD Italy, Via Vitorchiano 151, 00189 Rome, Italy; francesca.senese@merck.com (F.S.); gian.marco.prandi@merck.com (G.M.P.); 6Pedianet Project, 35138 Padua, Italy; a.scamarcia@sosepe.com (A.S.); l.cantarutti@sosepe.com (L.C.)

**Keywords:** pneumonia, invasive disease, non-invasive pneumonia, children, clinical burden, Italy

## Abstract

Despite advances in preventative interventions, invasive pneumococcal disease and pneumonia cause significant morbidity and mortality in children. We studied the annual incidence of pneumococcal-specific and syndromic invasive disease and non-invasive pneumonia in children <15 years of age during the early (2010–2013) and late (2014–2017) 13-valent pneumococcal conjugate vaccine (PCV13) periods in Veneto, Italy. In this retrospective observational study, pneumococcal-specific and syndromic invasive disease and non-invasive pneumonia cases were identified from several sources, including the Pedianet database. Interrupted time series analysis and Mann–Kendall tests were conducted to explore trends in incidence rates (IRs). Among 72,570 patients <15 years of age between 2010–2017, 88 episodes of pneumococcal-specific and syndromic invasive disease and 3926 episodes of non-invasive pneumonia were reported. Overall IR of pneumococcal-specific and syndromic invasive disease was 0.4/1000 person-years and did not change significantly (*p* = 0.46) throughout the study. Overall IR of non-invasive pneumonia was 10/1000 person-years and decreased significantly (−0.64, *p* = 0.026) over the study period. Following PCV13 introduction, the IRs of non-invasive pneumonia in children <15 years of age declined significantly, with no significant change in the IRs of pneumococcal-specific and syndromic invasive disease. There is a continuing clinical burden associated with pediatric pneumococcal diseases in Veneto, Italy.

## 1. Introduction

Despite advances in preventative interventions in recent years, invasive pneumococcal disease (IPD) and pneumonia continue to cause significant morbidity and mortality in children [1,2], with *Streptococcus pneumoniae* being a leading bacterial cause [3]. In Italy, the introduction of the 13-valent pneumococcal conjugate vaccine (PCV13) was associated with a reduction in the burden of IPD in children during the early years after it was introduced into the immunization schedule [4], although this was accompanied by a subsequent increase in IPD caused by non-vaccine serotypes [5,6]. 

Veneto is the fifth-largest region in Italy, with 9 local health authorities and >4.9 million inhabitants, of whom almost 700,000 are <15 years of age [7]. PCV13 replaced 7-valent PCV (PCV7) in the immunization schedule in Veneto in 2010. In Italy, PCV13 schedules for PCV7-naive children are three doses for infants (at 3, 5–6 and 11–13 months of age), two doses for children of 12–23 months of age (with a minimum gap of 2 months between the doses) and one dose for children 2–5 years of age [8]. However, from 2008–2017, mandatory vaccinations for pediatric patients were suspended in the region [9]; during this period, the vaccination coverage was in the range of 88–94% for the first dose, 89–94% for the second dose and 85–93% for the third dose [10]. The World Health Organization European Vaccine Action Plan for 2015–2020 recommends a vaccination coverage rate of ≥95% at a national level for all vaccines [11]. 

In Veneto, pneumonia-related hospitalizations in children 0–4 years of age decreased significantly (−9.3%, *p* < 0.01) between the early (2004–2005) and late (2006–2010) PCV7 periods; a further decrease in pneumonia-related hospitalizations was observed after the introduction of PCV13 (−19.7%) [12]. However, there are no recent data on the burden of IPD and pneumonia in the late PCV13 period. New vaccines are currently under development for the prevention of pneumococcal disease [13,14]. These investigational vaccines contain all serotypes in the currently licensed PCV13, as well as additional serotypes. To better understand the current burden of pneumococcal disease and the potential value of new vaccines in the Veneto region, it is important to quantify the time trends of IPD and pneumonia following the introduction of PCV13 and the residual burden that remains prior to the introduction of higher valent PCVs. This study assesses the annual incidence of pneumococcal-specific and syndromic invasive disease, as well as non-invasive pneumonia, following PCV13 introduction in Veneto in children <15 years of age, and compares the time trends in the early (2010–2013) and late (2014–2017) PCV13 periods. 

## 2. Methods

### 2.1. Study Design

This retrospective observational study was conducted using data from several sources. Among these was the Pedianet database [15], which is an organized network of 130 family physicians collecting electronic patient records for epidemiological and clinical research purposes. Data generated during routine patient care are collected and handled anonymously using a common software (JuniorBit®) (So.Se.Pe. s.r.l., Via G. Medici 9/A, 35138 Padua, Itlay), in compliance with Italian regulations, and stored under a unique numerical identifier. The Pedianet database records patient demographic characteristics, prescriptions, vaccinations, diagnoses and referrals to specialists and hospitals. Inclusion in the Pedianet database is voluntary.

The Veneto regional hospitalization and emergency room (ER) database, a centralized database that collects data on prescriptions, specialist visits, ER visits and hospitalizations for administrative and reimbursement purposes, was also used. This database is populated by various local health authorities and hospital trusts for administrative and reimbursement purposes. Data from the hospitalization and ER database can be linked to the Pedianet data through the Fascicolo Sanitario Project for all patients whose parent/legal guardian provides informed consent.

All pneumococcal-specific and syndromic invasive disease and non-invasive pneumonia-related visits and hospitalizations in patients <15 years of age from 1 January 2010 to 31 December 2017 were identified in the linked Pedianet and hospitalization and ER databases using International Classification of Diseases, Ninth Revision, Clinical Modification (ICD-9-CM) codes (Table 1). Additional free-text algorithms (““*olmon*””, ““*bp*””, ““*bn*””, ““*bm*””) were used to capture common terms used by physicians to describe pneumonia. Cases of invasive disease that were either specifically attributed to *S. pneumoniae*, or without known pathogen, but presenting with symptoms that matched conditions in which *S. pneumoniae* is known to have a causative role (such as bacterial meningitis or unspecified sepsis), were included. Non-invasive pneumonia included pneumonia caused by any pathogen. All cases were manually verified by a clinical data manager. 

### 2.2. Data Analysis

An episode could comprise 1 or more related hospitalizations, ER visits or family physician visits. A gap of 90 days between visits or hospitalizations with the identified diagnosis codes defined the start of a new episode. Episodes that crossed calendar years were assigned to the year in which the episode began.

Annual incidence rates (IRs) were defined as numbers of episodes per 1000 person-years, and 95% confidence intervals (CIs) were calculated using the following formula: ($IR\;\pm 1.96*\sqrt{\frac{events}{person-years^{2}\;}}*1000)$. Person-years were accumulated from the date at which a patient is registered in the Pedianet database (almost always coinciding with the date of birth) until death, migration or end of follow-up (31 December 2017). Interrupted time series (ITS) analysis was conducted using a negative binomial regression model to compare trends in annual IRs in the early (2010–2013) and late (2014–2017) PCV13 periods. Adjusted incidence rate ratios and 95% CIs were estimated for change in levels (i.e., an immediate change in the IRs from the previous period) and changes in trend (i.e., a gradual yearly change in the current IRs over time compared with the trends in the previous period) between the early and late PCV13 periods. The Mann–Kendall test was performed to determine whether the trend in annual IRs was monotonic.

## 3. Results

A total of 72,570 patients <15 years of age in the Veneto region were enrolled in the Pedianet database during the study period (2010–2017). In total, 88 episodes of pneumococcal-specific and syndromic invasive disease and 3926 episodes of non-invasive pneumonia were identified during the analysis period (Table 2). Among the invasive disease cases, 94% were attributable to syndromic invasive disease codes. 

The overall IR of pneumococcal-specific and syndromic invasive disease during the analysis period was 0.4 cases per 1000 person-years (95% CI 0.3–0.5) for children <15 years of age (Table 2). The annual incidence of pneumococcal-specific and syndromic invasive disease decreased from 0.4 cases per 1000 person-years (95% CI 0.2–0.6) in 2010 to 0.3 cases per 1000 person-years (95% CI 0.1–0.6) in 2017. Although the incidence of pneumococcal-specific and syndromic invasive disease was numerically lower in 2017 than in 2010, the ITS analysis did not detect a significant gradual change in either the early or late PCV13 periods, or a significant immediate change between the two periods (Table 3 and Figure 1). The Mann–Kendall test showed a non-significant monotonic decreasing trend (*p* = 0.46) over the study period. 

The demographic characteristics and risk factors for non-invasive pneumonia in children <15 years of age are shown in Table 4. A larger proportion of non-invasive pneumonia episodes occurred in males (52.4%) than in females (47.6%; Table 4). The largest proportion of pneumonia episodes were in children living in cities with a population range of 10,000–60,000 inhabitants (45.7%).

In all children <15 years of age, the mean annual IR of non-invasive pneumonia was 10 per 1000 person-years (95% CI 9–10). The IRs of non-invasive pneumonia decreased from 14 per 1000 person-years (95% CI 13–15), with 599 episodes in 2010, to 5 per 1000 person-years (95% CI 5–6) with 276 episodes in 2017 (Table 2). The IRs of non-invasive pneumonia were the highest in children 2–4 years of age across both the early and late post-PCV13 periods, with IRs at least twice that of children <2 years of age and up to 4 times the IRs in children 5–14 years of age (Table 5 and Figure 2). 

The results of the ITS analysis did not detect a significant change in the non-invasive pneumonia incidence in the early PCV13 period (Table 3). An immediate increase of 6.87% (*p* = 0.091) was observed between the early and late PCV13 intervention periods (Table 3). In the late PCV13 period, the IRs gradually decreased by 2.8% (*p* = 0.003) per year (Table 3 and Figure 2). The Mann–Kendall test showed a significant monotonic decreasing trend (*p* = 0.026) over the study period. 

When stratified by age, similar significant trends in non-invasive pneumonia incidence were observed for children in the 2–4 years and 5–14 years age groups, with IRs decreasing by 3.5% (*p* < 0.001) and 2.6% (*p* = 0.027) per year in the late PCV13 period, respectively (Figure 2). In children <2 years of age, no significant trend was detected (Figure 2). 

In all 3 age groups, there was an unexpected rise in the IRs of non-invasive pneumonia in 2014, resulting in the immediate increase between the early and late PCV13 periods. Further analysis of the monthly IRs of non-invasive pneumonia revealed that there was a peak in pneumonia episodes in the spring and summer months of 2014 in the Veneto region, which was not evident in any of the preceding or subsequent years (Figure 3). 

## 4. Discussion

In this study, the burden of pneumococcal-specific and syndromic invasive diseases did not change significantly following the introduction of PCV13, during 2010–2017. The burden of non-invasive pneumonia decreased, with the most benefits observed in older children. When the time trend of non-invasive pneumonia incidence was analyzed, a significant decrease in the late PCV13 period was observed. 

Although we observed a numerical change in the IRs of pneumococcal-specific and syndromic disease, we did not observe a significant decrease in the early or late PCV13 periods. The IRs of pneumococcal-specific and syndromic disease are similar to the IPD incidence previously observed in the Veneto region during 2011–2014 using the regional voluntary surveillance system [16]. The lack of a significant decrease may be due to the inclusion of pneumococcal-specific as well as syndromic disease, which may have been caused by bacterial and viral pathogens other than *S. pneumoniae* and may have obscured vaccine effects. Our study did not assess pneumococcal-specific invasive disease separately, as the number of cases was too small. However, in studies that assessed the incidence of invasive disease caused by *S. pneumoniae* specifically, PCV13 and PCV10 lowered the incidence of IPD [17,18,19]. In addition, the reduction in incidence of the vaccine-type IPD associated with PCVs was accompanied by a subsequent increase in IPD caused by non-vaccine serotypes [5,6]. This increase in non-vaccine-type IPD may be another reason for the persisting burden of pneumococcal-specific and syndromic diseases.

The reduction in incidence of non-invasive pneumonia observed in this study is consistent with previous studies conducted in pediatric populations of other countries following the introduction of pneumococcal vaccines to their national immunization programs [12,20,21,22]. A recent study conducted in the United Kingdom reported up to 34% incidence reduction in pneumonia of unspecified cause in children <2 years of age after the introduction of PCVs [22]. In the United Kingdom, PCV7 and PCV13 were associated with a 37% reduction in the incidence of all-cause pneumonia in children <10 years of age during 2002–2012 [20]. 

In our study, we noted an unexpected increase in the IRs of non-invasive pneumonia in the spring and summer months of 2014. The increase was more notable in children 2–4 and 5–15 years of age. It is possible that the increased incidence of pneumonia in 2014 seen in our study was caused by another pathogen, such as *Mycoplasma pneumoniae*. *M. pneumoniae* is a common cause of bacterial pneumonia in countries that have implemented successful PCV programs. It is detected more often in older children [23]. In Germany, Greece, Ireland and Slovenia, *M. pneumoniae* was identified as a common cause of pneumonia among children 5–14 years of age, although the pathogen was found in all age groups [24]. This notion was corroborated in a population-based surveillance study conducted in the US during 2010–2012. In this study, *M. pneumoniae* was detected less often in children <2 years of age compared with those >2 years of age and was more common than *S. pneumoniae* in the 2–4, 5–9 and 10–17 years of age groups [25]. We were unable to identify which pathogens were potentially responsible for the increase seen in our study because of the limited use of rapid tests and laboratory tests in the primary care settings and, therefore, the primary cause of the increase in non-invasive pneumonia can only be speculated. 

This study linked different sources of data at the patient level, including hospitalizations, ER visits, outpatient primary and specialty care visits and outpatient prescriptions. The strengths of our study include its size and representative coverage of pediatric patients; however, this study is limited by its retrospective nature. This study included both invasive diseases specifically attributed to *S. pneumoniae* and those without an identified pathogen but presenting with symptoms that matched conditions caused by *S. pneumoniae.* It is possible that this combination of cases may result in bias. Moreover, our study relied on ICD-9-CM codes of uncertain accuracy and disease etiology. Additional studies using *S. pneumoniae*-specific data may provide further information on the effects of pediatric pneumococcal vaccination. Increasing vaccine awareness and expanding the serotype coverage of next-generation vaccines to account for serotype shifts associated with pneumococcal disease are critical to reduce the overall burden of pneumococcal disease.

## 5. Conclusions

This study did not demonstrate a significant decline in pneumococcal-specific and syndromic invasive disease. However, we found a decrease in the IRs of non-invasive pneumonia in children <15 years of age following the introduction of PCV13, with the greatest benefit seen in older children. Our retrospective study indicates that there may still be a residual burden of pediatric pneumococcal diseases in the Veneto region of Italy. The extent to which future PCVs impact the burden of IPD and non-invasive pneumonia will depend on the proportion of diseases caused by *S. pneumoniae* and by vaccine-type serotypes.

## Figures and Tables

**Figure 1 children-09-00657-f001:**
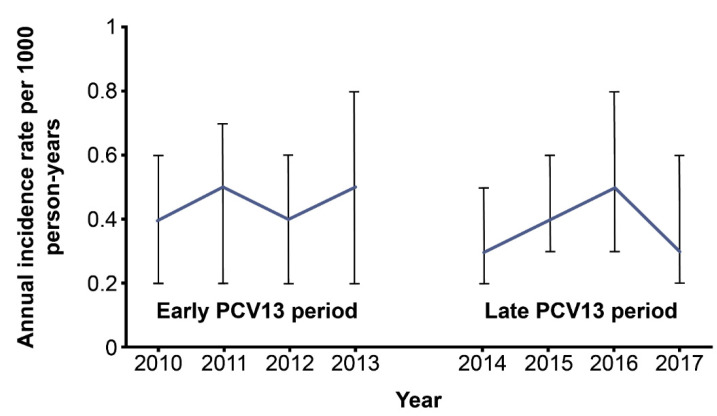
ITS trend for the mean annual pneumococcal-specific and syndromic invasive disease incidence in children. ITS, interrupted time series; PCV13, 13-valent pneumococcal conjugate vaccine. Error bars indicate 95% confidence intervals.

**Figure 2 children-09-00657-f002:**
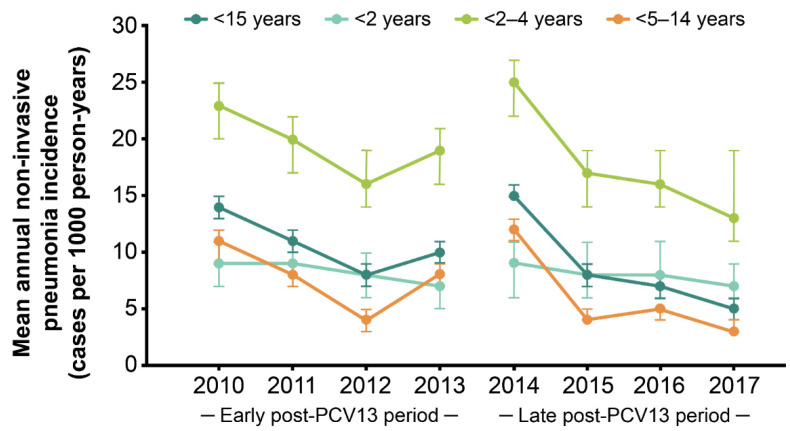
Changes in annual non-invasive pneumonia incidence following the introduction of PCV13. PCV13, 13-valent pneumococcal conjugate vaccine. Error bars indicate 95% confidence intervals.

**Figure 3 children-09-00657-f003:**
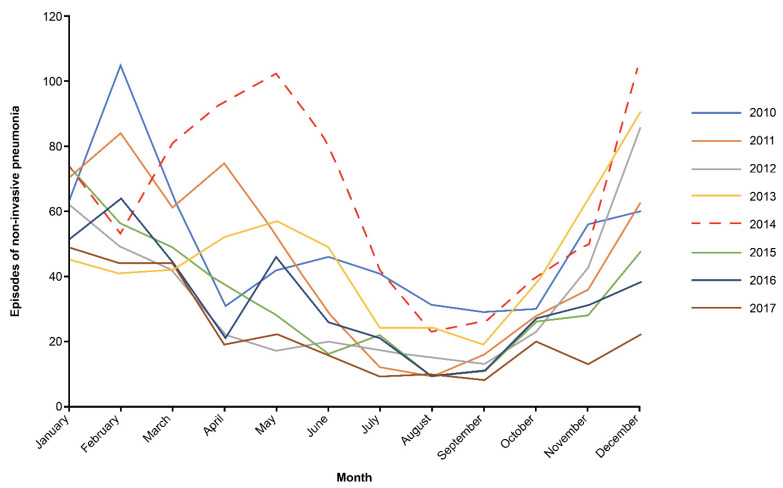
Monthly distribution of non-invasive pneumonia episodes in 2014 compared with other years.

**Table 1 children-09-00657-t001:** Search terms and diagnoses of pneumonia.

Term	Description
Non-invasive pneumonia	
ICD-9-CM code 480	Viral pneumonia
ICD-9-CM code 481	Pneumococcal pneumonia
ICD-9-CM code 482	Other bacterial pneumonia
ICD-9-CM code 483	Pneumonia due to another specified organism
ICD-9-CM code 484	Pneumonia in infectious diseases classified elsewhere
ICD-9-CM code 485	Bronchopneumonia, organism unspecified
ICD-9-CM code 486	Pneumonia, organism unspecified
ICD-9-CM code 487	Influenza

ICD-9-CM, International Classification of Diseases, Ninth Revision, Clinical Modification.

**Table 2 children-09-00657-t002:** Mean annual IR of invasive pneumococcal disease and non-invasive pneumonia in children.

Year	Pneumococcal-Specific and Syndromic Invasive Disease			Non-Invasive Pneumonia		
	Episodes, N	Person-Years, N	Mean Annual IR, Episodes per 1000 Person-Years (95% CI)	Episodes, N	Person-Years, N	Mean Annual IR, Episodes per 1000 Person-Years (95% CI)
2010	12	29,966	0.4 (0.2–0.6)	599	43,692	14 (13–15)
2011	14	29,990	0.5 (0.2–0.7)	534	47,610	11 (10–12)
2012	11	29,941	0.4 (0.2–0.6)	408	50,643	8 (7–9)
2013	15	30,039	0.5 (0.2–0.8)	545	52,192	10 (10–11)
2014	10	30,294	0.3 (0.1–0.5)	772	52,806	15 (14–16)
2015	9	22,997	0.4 (0.1–0.6)	403	52,392	8 (7–8)
2016	11	22,194	0.5 (0.2–0.8)	389	52,184	7 (7–8)
2017	6	19,223	0.3 (0.1–0.6)	276	51,348	5 (5–6)
Overall	88	214,644	0.4 (0.3–0.5)	3926	402,868 ^a^	10 (9–10)

CI, confidence interval; IR, incidence rate; ^a^ Numbers may not add up due to rounding.

**Table 3 children-09-00657-t003:** ITS analysis of annual IRs of invasive pneumococcal disease and non-invasive pneumonia.

	Early PCV13 Period (2010–2013)		Late PCV13 Period (2014–2017)	
	Coefficient	*p*-Value	Coefficient	*p*-Value
Pneumococcal-specific and syndromic invasive disease				
Trend	1.97	0.63	0.50	0.90
Change in trend	NA	NA	−1.47	0.80
Change in level ^a^	NA	NA	−9.34	0.50
Non-invasive pneumonia				
Trend	−1.13	0.237	−2.80	0.003
Change in trend	NA	NA	−1.50	0.319
Change in level ^a^	NA	NA	6.87	0.091

IR, incidence rate; ITS, interrupted time series; NA, not applicable; PCV13, 13-valent pneumococcal conjugate vaccine; ^a^ Immediate effect between the early and late post-PCV13 periods.

**Table 4 children-09-00657-t004:** Demographic characteristics and risk factors for non-invasive pneumonia in children.

	Non-Invasive Pneumonia (3926)
Characteristic/Risk Factor	*N* (%)
Male	2056 (52.4)
Female	1870 (47.6)
Urbanicity	
<10,000 inhabitants	896 (22.8)
10,000–60,000 inhabitants	1794 (45.7)
65,000–500,000 inhabitants	1236 (31.5)
Preterm birth	225 (5.7)
Underlying disease	
Asthma	35 (0.9)
Chronic heart disease	23 (0.6)
Chronic lung disease	4 (0.1)
Chronic renal failure	2 (0.1)

**Table 5 children-09-00657-t005:** Annual incidence estimates for non-invasive pneumonia stratified by age.

	<2 Years			2–4 Years			5–14 Years		
Year	Episodes, N	Person-Years, N	Mean Annual IR, Episodes per 1000 Person-Years (95% CI)	Episodes, N	Person-Years, N	Mean Annual IR, Episodes per 1000 Person-Years (95% CI)	Episodes, N	Person-Years, N	Mean Annual IR, Episodes per 1000 Person-Years (95% CI)
2010	72	8397	9 (7–11)	279	12,339	23 (20–25)	248	22,956	11 (9–12)
2011	76	8687	9 (7–11)	253	12,846	20 (17–22)	205	26,077	8 (7–9)
2012	70	8475	8 (6–10)	217	13,289	16 (14–19)	121	28,879	4 (3–5)
2013	53	7758	7 (5–9)	249	13,437	19 (16–21)	243	30,998	8 (7–9)
2014	58	6747	9 (6–11)	325	13,207	25 (22–27)	389	32,851	12 (11–13)
2015	46	5461	8 (6–11)	205	12,405	17 (14–19)	152	34,526	4 (4–5)
2016	44	5192	8 (6–11)	182	11,145	16 (14–19)	163	35,847	5 (4–5)
2017	37	5159	7 (5–9)	124	9630	13 (11–15)	115	36,559	3 (3–4)

CI, confidence interval; IR, incidence rate.

## Data Availability

The data used in this study cannot be made available in the manuscript; the supplemental files are in a public repository due to Italian data protection laws. The anonymized datasets generated during and/or analyzed during the current study can be provided upon reasonable request, from the corresponding author, after written approval by the Internal Scientific Committee (info@pedianet.it).

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
