# Peer review of "A Retrospective Analysis to Estimate the Burden of Invasive Pneumococcal Disease and Non-Invasive Pneumonia in Children <15 Years of Age in the Veneto Region, Italy"

_children, 2022, doi:10.3390/children9050657_

Round 1
Reviewer 1 Report
This is an interesting report on epidemiological data about pneumonia after introduction of PCV13.
I think study design description must be improved
Data presented comes from different sources. This could be a limit of this study.
Cases of invasive disease presenting with symptoms that matched conditions in which S. Pneumoniae is know to have a causative role were included but this can be considered as a bias
No data are presented about clinical or radiological diagnosis (lung ultrasound or chest Xray) of non invasive pneumonia and about treatment and hospital admission rate
In the discussion, the authors speculate that the increased incidence of pneumonia reported in 2014 could be due to other pathogens, but it is difficult to prove without etiological investigations. This limit should be emphasized more.
The authors assume that an increase in cases of Mycoplasma pneumonia may justify this finding, however it should be emphasized that Mycoplasma infection no longer recognizes a greater prevalence in older children.
Author Response
I think study design description must be improved.
We have added more detail to describe the study design. This is a retrospective database analysis that aims to assess the incidence rate of invasive IPD and non-invasive pneumonia in children.
Line: 75-109.
Data presented comes from different sources. This could be a limit of this study.
We believe that the larger number of sources can be considered as a strength and a limitation. It is a strength since it allows us to include more cases into the data set, however it can be a limitation as all sources were different and reporting parameters may have been different between these sources. Therefore we have not listed it as a strength or a limitation, but have clarified that different sources were used.
Line: 226-227.
Cases of invasive disease presenting with symptoms that matched conditions in which S. Pneumoniae is known to have a causative role were included but this can be considered as a bias.
We agree and have clarified that this could be a limitation of the study in the discussion section.
Line: 229-232.
No data are presented about clinical or radiological diagnosis (lung ultrasound or chest Xray) of non-invasive pneumonia and about treatment and hospital admission rate.
The primary aim of the study was to assess the burden, in terms of annual incidence, of pneumococcal-specific and syndromic invasive disease, as well as non-invasive pneumonia, following PCV13 introduction in Veneto in children 15 years of age, and compared the time trends in the early (2010–2013) and late (2014–2017) PCV13 periods. We were not interested in data on clinical or radiological diagnosis nor in the management of the cases.
In the discussion, the authors speculate that the increased incidence of pneumonia reported in 2014 could be due to other pathogens, but it is difficult to prove without etiological investigations. This limit should be emphasized more. The authors assume that an increase in cases of Mycoplasma pneumonia may justify this finding, however it should be emphasized that Mycoplasma infection no longer recognizes a greater prevalence in older children.
- We were unable to conduct etiological investigations to identify which pathogens were potentially responsible for the increase seen in our study because of the limited use of rapid tests and laboratory tests in the primary care settings. This is a limitation of the retrospective study design.
Although we agree that the prevalence of M. pneumoniae among older children has declined during the COVID-19 era, there is a paucity of bacterial co-infections seen during this era. Our study refers to data between the 2012–2017 period when its prevalence was high among school-age children.
Line: 221-225.
Reviewer 2 Report
Main comment
Although the study is well conducted, my biggest concern is the source of the data (Pedianet and ER databases) used for the study. Compared to the data in the article by Baldovin T et. Al (A Surveillance System for Invasive Pneumococcal Disease in Northeast Italy. Ann Ig. 2016;28:15.) there appears to be a significant underreporting bias for invasive pneumococcal disease. This could mean that, to a large extent, the results obtained in the analysis of these data were not completely reliable and, consequently, the conclusions drawn by the authors might not be correct.
Minor comments
Table 2. Please, explain in Material and Methods how was the number of “Person-years, n” obtained for invasive and non-invasive disease. Why are the numbers of person-year not the same for invasive and non-invasive diseases? Besides, in Line 112 it is stated that there were 72,570 patients <15 years of age in the region.
Line 115-116. Among the invasive disease cases, 94% were attributable to syndromic invasive disease codes.
If there were a total of 88 episodes of invasive disease, this means that only around 5 cases (6%) in 8 years were due to pneumococcal-specific invasive cases. However, in a study of Baldovin. et al, between 2011 and 2014 they found 47 cases of IPD confirmed by culture in children under 15 years old in the Veneto region (around 1,7 cases /100.000). These data undermine the credibility of the source from which the data were obtained for the study.
Baldovin T, Lazzari R, Russo F, Bertoncello C, Buja A, Furlan P, Cocchio S, Palù G, Baldo V. A surveillance system of Invasive Pneumococcal Disease in North-Eastern Italy. Ann Ig. 2016 Jan-Feb;28(1):15-24.
Discussion. It is strange that the above-mentioned article of BaldovinT et al. on IPD in the Veneto between 2007-2014 is not mentioned in the Discussion.
Lines 204-205. It is possible that the increased incidence of pneumonia in 2014 seen in our study was caused by another pathogen, such as Mycoplasma pneumoniae.
Although the authors provide data from other countries, this statement on Mycoplasma infection is highly speculative without showing own (or at least Italian) data.
Lines 31-32 & Line 224. There remains a substantial clinical burden of pediatric pneumococcal diseases in the Veneto region of Italy.
Although the sentence is probably true, it has not been demonstrated in the results, as no results of specific pneumococcal diseases have been provided (nearly no cases of invasive disease and nonspecific study on pneumococcal pneumonia).
Author Response
Although the study is well conducted, my biggest concern is the source of the data (Pedianet and ER databases) used for the study. Compared to the data in the article by Baldovin T et. Al (A Surveillance System for Invasive Pneumococcal Disease in Northeast Italy. Ann Ig. 2016;28:15.) there appears to be a significant underreporting bias for invasive pneumococcal disease. This could mean that, to a large extent, the results obtained in the analysis of these data were not completely reliable and, consequently, the conclusions drawn by the authors might not be correct.
Baldovin et al. was a prospective study based on a regional surveillance reporting system. We understand the limitations of retrospective studies from medical databases such as Pedianet and ER databases, but there is still value in interrogating such databases by using valid analytical processes. Moreover, we confirm that our IPD incidence is similar to the IPD notification rate mentioned in Baldovin et al. (i.e. 2.32/100,000 vs 2.15 (not 1.7) cases /100,000).
Line: 190-192.
Table 2. Please, explain in Material and Methods how was the number of “Person-years, n” obtained for invasive and non-invasive disease. Why are the numbers of person-year not the same for invasive and non-invasive diseases? Besides, in Line 112 it is stated that there were 72,570 patients <15 years of age in the region.
We have included an explanation of how this was obtained in the Methods sections. However, since we did not have the access to the ER database for all children included in the cohort, but just for a part, we assessed the invasive disease only among these children. We have already specified in the manuscript that “Data from the hospitalization and ER database can be linked to Pedianet data through the Fascicolo Sanitario Project for all patients whose parent/legal guardian provides informed consent.”
Line: 107-109.
Line 115-116. Among the invasive disease cases, 94% were attributable to syndromic invasive disease codes.
If there were a total of 88 episodes of invasive disease, this means that only around 5 cases (6%) in 8 years were due to pneumococcal-specific invasive cases. However, in a study of Baldovin. et al, between 2011 and 2014 they found 47 cases of IPD confirmed by culture in children under 15 years old in the Veneto region (around 1,7 cases /100.000). These data undermine the credibility of the source from which the data were obtained for the study.
Baldovin T, Lazzari R, Russo F, Bertoncello C, Buja A, Furlan P, Cocchio S, Palù G, Baldo V. A surveillance system of Invasive Pneumococcal Disease in North-Eastern Italy. Ann Ig. 2016 Jan-Feb;28(1):15-24.
- We thank the reviewer for the comment. We confirm that our IPD incidence is similar to the IPD notification rate mentioned in Baldovin et al. (i.e. 2.32/100,000 vs 2.15 (not 1.7) cases /100,000 inhabitants). We acknowledge that the units used are different (i.e. incidence rate vs IPD notification rate), hence it is difficult to compare results. However, following this, we do not believe that the credibility of our study is undermined by our data source.
Line: 190-192.
Discussion. It is strange that the above-mentioned article of BaldovinT et al. on IPD in the Veneto between 2007-2014 is not mentioned in the Discussion.
Lines 204-205. It is possible that the increased incidence of pneumonia in 2014 seen in our study was caused by another pathogen, such as Mycoplasma pneumoniae.
Although the authors provide data from other countries, this statement on Mycoplasma infection is highly speculative without showing own (or at least Italian) data.
- We have added a sentence to mention the similarity of our results with Baldovin et al. in the discussion section.
We were unable to conduct our own etiological investigations to identify the pathogens potentially responsible for the increase seen in our study because of the poor use of rapid tests and laboratory tests in the primary care settings. Our study refers to data between the 2012–2017 period and there is a paucity of Italian data on M. pneumoniae infections during this time period. We have therefore provided other European data to support our observation and clarified that the causative organism can only be speculated due to the lack of data from epidemiological studies.
Line: 190–192 and 218–225.
Lines 31-32 & Line 224. There remains a substantial clinical burden of pediatric pneumococcal diseases in the Veneto region of Italy.
Although the sentence is probably true, it has not been demonstrated in the results, as no results of specific pneumococcal diseases have been provided (nearly no cases of invasive disease and nonspecific study on pneumococcal pneumonia).
We have softened the conclusion statement to align with the data presented and mentioned the residual burden that continues to exist with pneumonia and IPD.
Line: 242–243.
Reviewer 3 Report
This study describes a large pediatric dataset covering both a broad geographic region and long time frame, evaluating the impact of PCV13 on invasive and non-invasive pneumococccal infections over time. While the results did not show a significant decline, the authors do a nice job framing the results among what is already known in the field. The authors also appropriately acknowledge the limitations of the work and the analytic plan is sound. The manuscript could be strengthened by (1) including some assessment of medical complexity in the patient population, if possible, to determine if factors other than age factor into IR; and (2) expanding discussion of next steps to move this field forward.
Author Response
This study describes a large pediatric dataset covering both a broad geographic region and long time frame, evaluating the impact of PCV13 on invasive and non-invasive pneumococccal infections over time. While the results did not show a significant decline, the authors do a nice job framing the results among what is already known in the field. The authors also appropriately acknowledge the limitations of the work and the analytic plan is sound. The manuscript could be strengthened by (1) including some assessment of medical complexity in the patient population, if possible, to determine if factors other than age factor into IR; and (2) expanding discussion of next steps to move this field forward.
We thank the reviewer for the feedback.
- The primary aim of the study was to assess the burden, in terms of annual incidence, of pneumococcal-specific and syndromic invasive disease, as well as non-invasive pneumonia, following PCV13 introduction in Veneto in children <15 years of age, and compared the time trends in the early (2010–2013) and late (2014–2017) PCV13 periods. We were not interested in assessment of medical complexities within the patient population.
- We have added a sentence to the discussion section to cover next steps in reducing pneumococcal disease burden.
Line: 235-237.
Round 2
Reviewer 2 Report
The authors have correctly addressed all issues raised by this reviewer.